# In Vitro and In Vivo Characterization of a New Strain of Mosquito Flavivirus Derived from *Culicoides*

**DOI:** 10.3390/v14061298

**Published:** 2022-06-14

**Authors:** Yi Huang, Hongqing Zhang, Xiaodan Li, Lu Zhao, Dirui Cai, Shunlong Wang, Nanjie Ren, Haixia Ma, Doudou Huang, Fei Wang, Zhiming Yuan, Bo Zhang, Han Xia

**Affiliations:** 1Key Laboratory of Special Pathogens and Biosafety, Wuhan Institute of Virology, Chinese Academy of Sciences, Wuhan 430071, China; huangyi1104451379@163.com (Y.H.); zhanghongqing24@163.com (H.Z.); lxd@live.cn (X.L.); paddlesaddle@foxmail.com (S.W.); nanjieren@outlook.com (N.R.); haixiam@wh.iov.cn (H.M.); huangdd@wh.iov.cn (D.H.); wangfei@wh.iov.cn (F.W.); yzm@wh.iov.cn (Z.Y.); 2University of Chinese Academy of Sciences, Beijing 100049, China; 3Westlake Disease Modeling Lab, Westlake Laboratory of Life Sciences and Biomedicine, Hangzhou 310024, China; zhaolu@westlake.edu.cn; 4School of Life Sciences, Westlake University, Hangzhou 310024, China; 5School of Life Sciences, Hubei University, Wuhan 430062, China; cai18520813369@126.com

**Keywords:** insect-specific virus, mosquito flavivirus, *Culicoides*, reverse genetic system, next-generation sequencing (NGS), Yunnan

## Abstract

Mosquito-specific flaviviruses comprise a group of insect-specific viruses with a single positive RNA, which can affect the duplication of mosquito-borne viruses and the life growth of mosquitoes, and which have the potential to be developed as a vaccine platform for mosquito-borne viruses. In this study, a strain of mosquito flavivirus (MFV) YN15-283-02 was detected in *Culicoides* collected from Yunnan, China. The isolation of the purified MFV YN15-283-02 from cell culture failed, and the virus was then rescued by an infectious clone. To study the biological features of MFV YN15-283-02 in vitro and in vivo, electron microscopy, phylogenetic tree, and viral growth kinetic analyses were performed in both cell lines and mosquitoes. The rescued MFV (rMFV) YN15-283-02 duplicated and reached a peak in C6/36 cells at 6 d.p.i. with approximately 2 × 10^6^ RNA copies/μL (RNA to cell ratio of 0.1), but without displaying a cytopathic effect. In addition, the infection rate for the rMFV in *Ae.*
*aegypti* show a low level in both larvae (≤15%) and adult mosquitoes (≤12%).

## 1. Introduction

Over the past several decades, with the advent of next-generation sequencing (NGS) technology, various insect-specific viruses (ISVs) have been discovered globally [1]. Hundreds of ISVs have been isolated and documented [2,3]. ISVs are distributed in families that mainly include *Flaviviridae*, *Togaviridae*, *Mensoniviridae*, and *Rhabdoviridae* [4]. ISVs have been explored as biological control agents in insect, vaccine and diagnosis platforms for arbovirus [5,6,7]. In addition, studies revealed that ISVs can affect arbovirus replication both in cell cultures and arthropods [8]. *Culicoides*, a biting midge, one of the vectors of pathogens, can propagate arboviruses that include Akabane virus, Tinaroo virus, and Aino virus [9,10,11]. In the *Flavivirus* genus, West Nile virus has been detected in *C. arboricola*, *C. biguttatus*, and *C. stellifer*.

*Flaviviruses* can be divided into four groups including tick-borne flaviviruses (TBFs), mosquito-borne flaviviruses (MBFVs), classical insect-specific flaviviruses (cISFs), and dual-host affiliated insect-specific flaviviruses (dISFs) [12,13]. They are genera of viruses that are 40–60 nm diameter enveloped spheres that contain positive-strand RNA. The full-genome sequence comprises a single open reading frame flanked by a 5′ untranslated region (UTR) of 100 bp nucleotides and a 3′UTR of 400–700 bp nucleotides [14,15,16]. Polyprotein is cleaved by cellular and viral proteases to generate three structural proteins—capsid (C), pre-membrane/membrane (prM/M), and envelope (E) proteins—and seven non-structural functional proteins [17,18,19,20,21,22].

Recently, many studies have investigated the interaction of ISVs with arbovirus, the transmission pattern of ISVs, and the potential to biologically control ISVs in mosquitoes. For example, Espirito Santo virus superinfected with DENV could restrain the replication of DENV and the spreading ability of DENV in *Ae. aegypti*, declining vector competence [23]. In addition, the densovirus has potent larvicidal activity and is able to infect various mosquitoes such as genera *Aedes*, *Culex,* and *Culiseta* [24]. Intrathoracic inoculation is used to infect larvae with ISVs such as Culex flavivirus to simulate viral horizontal transmission [8,25].

A previous study discovered a novel mosquito flavivirus (MFV), LSFlaviV-A20-09, which was isolated from *Culex tritaeniorhynchus* in Yunnan, China, but they only reported the viral morphology and the phylogenetic tree based on the viral genome [26]. In this study, a new strain of MFV, YN15-283-02, was detected in the *Culicoides* homogenate, and next-generation sequencing (NGS) and electronic microscopy revealed two types of viruses, flaviviruses and orbiviruses, which were presented in this sample. Then, the MFV YN15-283-02 was rescued using a full-length infectious clone, since we could not achieve the purified isolation of it using a cell culture. Finally, the infection and growth characteristics of the rescued MFV YN15-283-02 in cell culture and *Ae. Aegypti* were investigated.

## 2. Materials and Methods

### 2.1. Cell Culture and Mosquitos

C6/36 cells (CRL-1600; American Type Culture Collection, *Manassas,* VA, USA) were grown in RPMI 1640 medium containing 10% fetal bovine serum (FBS) and 1% penicillin/streptomycin at 28 °C in an atmosphere of 5% CO_2._

*Ae. aegypti* (kindly provided by Prof. Qian Han from Hainan University, Haikou, China) eggs were hatched. The larvae were fed fish fodder in deionized water at a temperature of 28 °C and reared at 70–80% humidity and 12 h light. After a week, pupae that emerged were transferred to a 30 × 30 × 30 cm^3^ cage. The adult mosquitoes were fed 8% glucose to maintain growth after eclosion. To obtain the next generation, female adult mosquitoes were allowed to feed on horse blood through the Membrane Feeding System (Hemotek, UK).

### 2.2. Electron Microscopy

A cytopathic effect (CPE) was detected in a pool of *Culicoides* homogenate (YN15-283)-inoculated C6/36 cells [27]. The supernatant from P1 was inoculated in 8 T175 flasks and used for transmission electron microscopy (TEM).

To concentrate the virions, the pooled supernatants were centrifugated at 125 g at 4 °C for 10 min. The sediment of cell debris was discarded. Then, the supernatant was transferred to an SW32 ultracentrifuge tube containing a 20% wt/vol sucrose cushion. Ultracentrifugation was performed at 31,000× *g* and 4 °C for 3 h. The sediment was resuspended immediately in 100 μL of phosphate-buffered saline (PBS). The suspension was filtered through a 0.5 mL Amicon^®^ Ultra ultracentrifugal filter device. The collected virions were suspended in PBS, and drops were dispensed on Formvar-coated copper grids and stained with 2% uranyl acetate. The excess stain was removed in the usual manner, and TEM was performed using a model Tecnai G2 20 TWIN operating at 200 kV (FEI, Hillsboro, OR, USA).

### 2.3. RNA Extraction and qRT-PCR

For the cell culture sample, the supernatant of virus-infected cell culture was obtained. RNA was extracted from 140 μL of supernatant using the QIAamp viral RNA mini kit (52906; QIAGEN, Germantown, MD, USA).

For the mosquito sample, mosquito larvae, pupae, or adults were collected individually into a sterile tube with 3 mm ceramic beads (NovaSta, Xi’an, Shaanxi, China) and 300 μL of PBS, and then, they were homogenized in a tissue homogenizer (Servicebio, Wuhan, Hubei, China) for 30 s/50 Hz at 4 °C for 3 cycles. Each milled mosquito sample was centrifuged at 10,000× *g* for 10 min to remove the debris. Then, the RNA was extracted from 200 μL of supernatant in the above step using the Magnetic Virus RNA Extraction kit (NanoMagBio, Wuhan, Hubei, China), as described by the manufacturer.

The qRT-PCR primers for MFV were designed and checked using a BLAST search online to avoid nonspecific binding (https://blast.ncbi.nlm.nih.gov/Blast.cgi (accessed on May 2021)) and commercially synthesized using TSINGKE (Wuhan Branch, China) (Appendix A). qRT-PCR was conducted using PrimeScript™ One Step RT-PCR kit (Takara Bio, Shiga, Japan) in the thermocycler (CFX96™ Real-Time System; BIO-RAD, Hercules, CA, USA).

### 2.4. Next Generation Sequencing and Genome Assembly

RNA extracted from the supernatant of inoculated C6/36 cell culture was sequenced using the HiSeq 2500 platform (Illumina, CA, USA) by Nextomics Bioscience Co., Ltd. (Wuhan, China). The analysis of the sequencing data was performed with a self-designed Bioinformatics pipeline. Briefly, adapters and low-quality reads were trimmed from raw paired-end reads using Trim-galore v0.6.6 [28]. Then, the reads of host cells were removed with the genome of *Aedes Albopictus* (NCBI Assembly GCA_006496715.1) using bowtie2 v2.4.2 [29,30]. The cleaned reads were de novo assembled to contigs using Trinity v2.11.0 [31,32]. The contigs were aligned against the NCBI nonredundant nucleotide database (updated in March, 2021) with BLASTn for viral species classification [33]. Then, the cleaned reads were aligned to the acquired contig with the significant similarity to the reference virus (MFV LSFlaviV-A20-09) to calculate the sequencing depth using bowtie2 v2.4.2 and samtools v1.11. Afterward, the 5′ untranslated region (UTR) and 3′ UTR were amplified via the rapid amplification of cDNA ends (RACE) to compensate for the gaps in the terminal region. Eventually, the complete genome of the target virus was acquired and submitted to Genbank.

### 2.5. Phylogenetic Analysis

The representative flaviviruses were derived from 4 groups consisting of TBFs (6 viruses), MBFVs (14 viruses), cISFs (13 viruses), and dISFs (7 viruses), referring to the study of Blitvich et al. [13]. In addition, MFV YN15-283-02, isolated MFV LSFlaviV-A20-09 (NC_021069.1), and Culex theileri flavivirus (HE574574.1) were supplemented in the phylogenetic tree. The amino acid sequence of the complete polyprotein of each flavivirus was downloaded from GenBank using TBtools v1.096 (Appendix A) [33]. Multiple alignments of amino acid sequences were performed using MUSCLE v3.8.425 with default parameters (https://www.ebi.ac.uk/Tools/msa/muscle/ (accessed on 12 May 2022)). Phylogenetically informative sites were selected from the MUSCLE alignment result by using Gblocks v0.91b in PhyloSuite v1.1.16 [34]. The phylogenetic tree was constructed using IQ-TREE in PhyloSuite v1.1.16 with the maximum likelihood method of 5000-fold ultrafast bootstrap [35,36]. Afterward, the tree was modified using iTOL (https://itol.embl.de/upload.cgi (accessed on 12 May 2022)).

### 2.6. Construction of Infectious Clone and Virus Rescue

A schematic diagram of the construction of the full-length clone of MFV YN15-283-02 is shown (Figure 1). Briefly, five cDNA fragments (A to E) covering the full length of the virus genome were obtained via reverse transcription PCR using total RNA extracted from the supernatant. Fragment A contained a T7 promoter and sequences ranging from the 5′UTR to 1262 nt of the genome. Fragments B, C, and D contained the genome sequence from nucleotide positions 867 to 3260 nt, 3084 to 6628 nt, and 6400 to 7675 nt, respectively. Fragment E contained the sequence from 7438 nt to 3′UTR of the genome. Fragment A, B, C, D, and E were individually cloned into a low-copy-number vector, pACYC177, at the Sma I and Nhe I sites. Fragment A + B was generated by inserting fragment B into fragment A at the Spe I and Nhe I sites. Fragment C was introduced into fragment A + B at the Cla I and Nhe I sites, producing fragment A–C. Subsequently, fragment A–D was generated by incorporating fragment D into fragment A–C at the Mfe I and Nhe I sites. Ultimately, the infectious clone pACYC-MFV/YN15-283-02 containing a T7 promoter and the complete genome of virus was obtained via the assembly of fragment E into fragment A–D at the Kpn I and Nhe I sites.

To rescue MFV YN15-283-02, transcripts derived from the infectious clones were used to transfect C6/36 cells. The infectious clone was linearized by Nhe I and purified using phenol/chloroform extraction. MFV YN15-283-02 RNA was transcribed in vitro from the linearized product using the T7 mMEGAscript^®^ Kit (Ambion, TX, USA) according to the manufacturer’s protocols. The transcript (1 μg) was transfected into C6/36 cells with DMRIE-C reagent (Invitrogen, CA, USA). The supernatant of the transfected cells was obtained at different time points after transfection, aliquoted, and stored at −80 °C for use.

### 2.7. Viral Infection and Growth Characteristics in Cell Culture

The growth kinetics of the rescued MFV YN15-283-02 (rMFV YN15-283-02) was determined in BHK-21 vertebrate cells and C6/36 invertebrate cells. Briefly, cells were seeded into 35 mm dishes (6 × 10^5^ per dish). After cultivation at 28 °C for one day in an atmosphere of 5% CO_2_, the cells were incubated with virus at a ratio of viral RNA copies to cell number of 0.1 and 400. After incubation for 2 h, the cell supernatant was removed, washed three times with PBS, and replaced by fresh RPMI-1640 medium containing 2% FBS. At different time points post-infection, the cell supernatants were collected and stored at −80 °C for RNA extraction. The number of copies of genomic RNA was determined with a standard curve of the in vitro transcribed rMFV YN15-283-02 RNA.

### 2.8. Viral Infection and Growth Characters in Aedes Mosquitoes

For larvae mosquito infection via oral feeding [37,38], first-instar larvae were collected 48 h after egg hatching. The larvae were separated in a 6-well plate (*n* ≥ 20 per well) and exposed to 9 mL of an aqueous suspension of virus. After incubation for 48 h at 28 °C, larvae were washed three times with deionized water and then transferred to clean pans containing fresh water. The fourth-instar larvae, pupae, and adult mosquitoes (males and females) were harvested for virus detection.

For the adult mosquito infection, adult 5–8-day-old female mosquitoes were collected and starved for 24 h, and then fed a blood meal containing virus (horse blood:virus (*v*/*v*) = 1:1) for approximately 1 h using the Membrane Feeding System (Hemotek, Blackburn, UK). Fully engorged mosquitoes were selected and put into mosquito cartons. After 7, 10, and 14 days post-infection (d.p.i), the mosquitos were collected for virus detection. Statistical significance was determined using the Holm–Sidak method, with α = 0.05.

## 3. Results

### 3.1. Detection of Two Types of Viral Particles in Supernatant from Midge Homogenate-Inoculated Cell Culture

The supernatant of homogenate from one pool (YN15-283) of *Culicoides* spp. was inoculated into C6/36 monolayer cells, and CPE was observed after 5 days. Then, the supernatant for the cell culture of YN15-283 was concentrated via ultracentrifugation and observed using TEM. There were two types of spherical virions with diameters of approximately 30–40 nm and 70–80 nm (Figure 2A). The morphology of the rMFV YN15-283-02 was also observed using TEM, which was shown to be similar to that of wild viral particles (Figure 2B).

### 3.2. Genome Sequence and Phylogenetic Analysis of Mosquito Flavivirus YN15-283-02

Sequencing data of the inoculated C6/36 cell culture contained a total of 3,241,470 cleaned reads. After de novo assembly, 6724 contigs were acquired, and 1 contig (contig 1, 10,486 nt) was hit best to the MFV isolate LSFlaviV-A20-09, and 12 contigs (contig 2 to 12) matched to Tibet orbivirus through BLASTn (Table 1).

The sequencing depth and the predicted region of contig 1 are plotted in Figure 3A, which indicated contig 1 covered the full CDS region of MFV. Then, the gaps in the 5′ UTR and 3′ UTR were compensated by RACE, and finally, the complete genome of MFV YN15-283-02 (MZ821064.1) was acquired.

The phylogenetic tree was constructed based on the amino acid sequences of the polyprotein for MFV and the representative viruses from cISFs, dISFs, TBFs, and MBFVs. The result indicated that the MFV YN15-283-02 lay on the cluster of the cISFs, which was closest to the MFV LSFlaviV-A20-09 (Figure 3B).

### 3.3. Rescue of Mosquito Flavivirus YN15-283-02 and Growth Characteristics in C6/36 and BHK-21 Cell Lines

The full-length infectious clone pACYC-MFV/YN15-283-02 plasmid was obtained via the sequential assembly of the individual subclones using the unique restriction sites within the genome. YN15-283-02 RNA transcribed from linearized pACYC-MFV/YN15-283-02 was transfected into C6/36 cells, and the virus production was confirmed using RT-PCR detection specific to the MFV YN15-283-02 MFV RT-PCR products, which could be detected in celllular RNA for P0, P1, and P2, indicating that the infectious virus was successfully rescued by the infectious clone (Figure 4A).

When C6/36 cells were infected with rMFV YN15-283-02 at a ratio of viral RNA copies to cell number of 0.1, the viral RNAs significantly increased and reached a peak (2 × 10^6^ copies/μL) at 6 d.p.i. in the harvest supernatants. This indicates the efficient replication of the rMFV YN15-283-02 (Figure 4B) in C6/36. To further confirm the host specificity of rMFV, mammalian BHK-21 cells and mosquito C6/36 cells were infected with high doses of rMFV at a ratio of viral RNA copies to cell number of 400, and the result showed that rMFV cannot replicate in BHK cells (Figure 4C,D).

### 3.4. Infection and Growth Characteristics of Rescued Mosquito Flavivirus YN15-283-02 in Ae. aegypti Mosquito

The process of mosquito infection is presented in Figure 5A. In the larvae infection (Figure 5B), the infection rate in larvae, pupae, and F0 adults was 0%, 0%, and 5% in the low-dose infection group (6.18 × 10^4^ copies/μL) and 15%, 5%, and 10% in the high-dose infection group (2.53 × 10^7^ copies/μL). In the stage of larvae and pupae, there were significant differences in the viral loads (*p* = 0.01 and *p* < 0.001, respectively) between the high- and low-dose groups, but no differences were detected in the F0 adult mosquitoes.

Regarding adult infection, (Figure 5C), the infection rate at 7, 10, and 14 d.p.i was 12%, 8%, and 10% in the low-dose infection group (5.62 × 10^3^ copies/μL) and 10%, 0%, and 8% in the high-dose group (1.01 × 10^5^ copies/μL). In addition, there were no significant differences in the viral loads in adult mosquitos between the high- and low-dose groups.

Moreover, the viral loads in adult mosquitos via blood-feeding infection displayed differences in individuals in a range from 1.51 × 10^2^ to 1.02 × 10^5^ copies/μL, which contrasted with the viral loads in larvae, pupae, and F0 adults in a range from 1.26 × 10^2^ to 4.47 × 10^2^ copies/μL. These data reveal that the rMFV YN15-283-02 has low infection ability in *Ae.*
*aegypti* [39,40]. 

## 4. Discussion

Here, we detected, sequenced, and rescued a mosquito-specific flavivirus derived from *Culicoides* and studied the biological characteristics of virus infection both in vitro (cell culture) and in vivo (mosquitoes). In the natural environment, several ISVs co-exist with arbovirus [41]. To separate the MFV YN15-283-02 from Tibet obivirus (TIBOV) in the co-infection sample, we tried both the methods of infinitely diluted and plaque purification in C6/36 cells. Unfortunately, we failed to obtain purified isolates of MFV from the co-infection mixture. Therefore, the virus was rescued using an infectious clone, and rMFV was used for the infection in the cell cultures and mosquitoes. As the two viruses co-exist in nature, we could investigate the interaction between MFV and TIBOV in cells and mosquitoes in the next step.

Viruses could be detected or replicated both in midges and mosquitoes [9,10]. Several ISFVs affect arbovirus replication and transmission both in vivo and in vitro [39,40,42] and have potent larvicidal activity [24,43,44]. In this study, the MFV YN15-283-02 is an insect-specific flavivirus discovered from *Culicoides*. The MFV YN15-283-02 displayed high nucleotide identity (98.9%) with the isolated LSFlaviV-A20-09. Interestingly, the MFV YN15-283-02 was unable to produce CPE in C6/36 cells, in contrast to the isolated LSFlaviV-A20-09 [26,27]. In addition, the isolated MFV LSFlaviV-A20-09 was isolated from mosquito, no information was found as to whether this virus can infect and replicate in mosquitoes or other vectors. Here, we studied the infection characteristics of the rMFV YN15-283-02 in *Ae.*
*aegypti*, an important vector for many mosquito-borne arboviruses. The rMFV YN15-283-02 displayed a low infection capability in both larvae and adult mosquitoes, with no pathogenicity to *Ae.*
*aegypti*. In further study, experimental infection in different species of mosquitos and midges could be conducted to reveal the primary vector species for MFVs.

In addition, due to the lack of the CPE in the mosquito cell, a recombinant MFV could be constructed to carry a reporter gene which could help to track the virus easily. At the same time, the recombinant MFV could be used as a powerful tool in a study to identify the critical factor in the flavivirus genome and determine the species limitations and virus–host interaction.

The reverse genetics system is a powerful molecular tool for RNA virus research and vaccine development. Infectious clones of flaviviruses such as Zika virus (ZIKV) and West Nile virus (WNV) have been successfully constructed and rationally engineered for vaccine design [45,46]. As ISFVs can only infect arthropod vectors and not the vertebrate hosts, they represent identical viral vectors for vaccine development. Peters et al. constructed an infectious clone of Binjari virus (BinJV) and developed vaccine candidates BinJV-WNV-prME and BinJV-ZIKV-prME by substituting the prME of BinJV with those of WNV and ZIKV, which remained insect-specific and showed great immune response and protection against lethal challenges [47]. The method of constructing vaccine candidate strains based on reverse genetics is rapid, efficient, and targeted, meaning it has good potential to be applied in vaccine research.

## Figures and Tables

**Figure 1 viruses-14-01298-f001:**
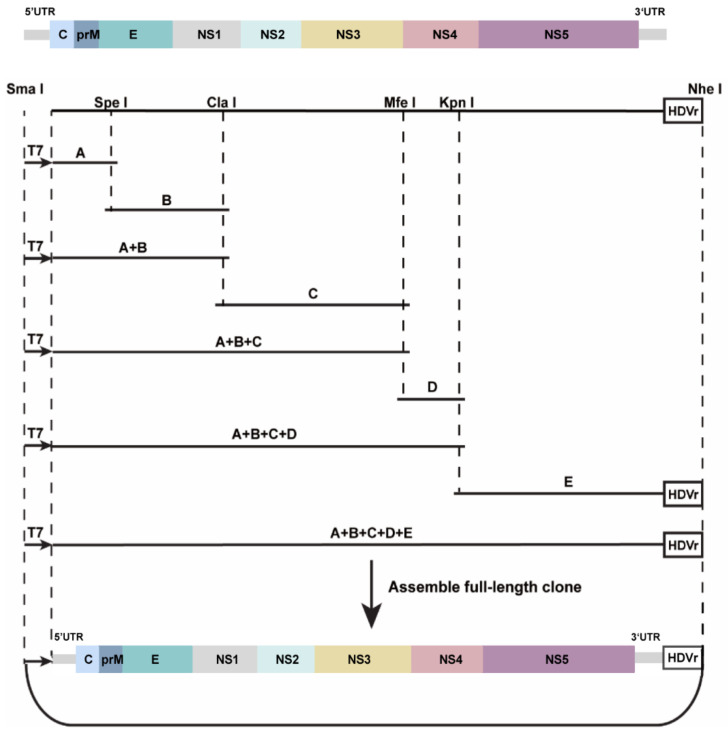
Schematic diagram of the cloning strategy to construct infectious clone for mosquito flavivirus strain YN15-283-02. Genome distribution and restriction sites used for genome assembly are shown. Five cDNA fragments (A–E) covering the complete MFV YN15-283-02 genome were amplified using RT-PCR. A T7 promoter was fused upstream of the 5′UTR in fragment A and an HDVr downstream of the 3′UTR in fragment E.

**Figure 2 viruses-14-01298-f002:**
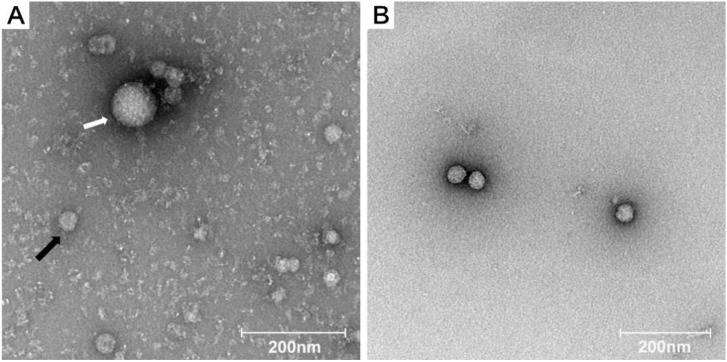
Electron microscopy of wild (**A**) and rescued (**B**) mosquito flavivirus. (**A**) Two types of virions were observed with diameters of 70–80 nm (white arrow) and 30–40 nm (black arrow). (**B**) The diameter of rMFV YN15-283-02 was around 30–40 nm. Scare bar = 200 nm.

**Figure 3 viruses-14-01298-f003:**
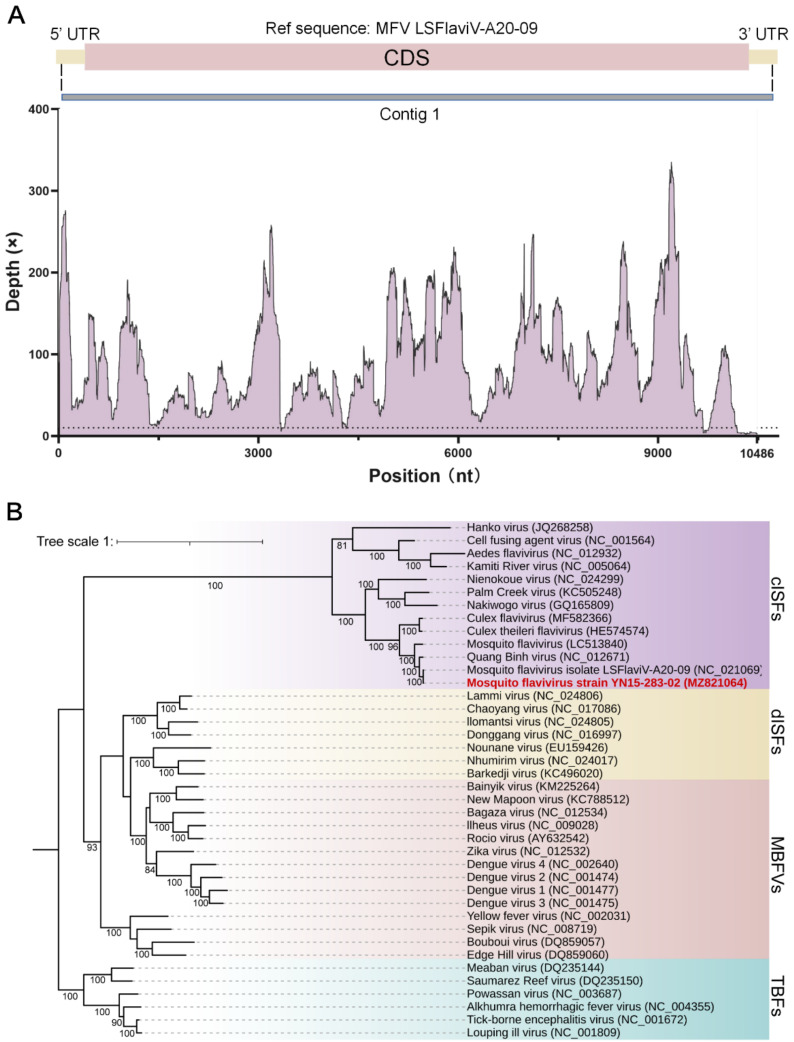
(**A**) The sequencing depth and the coverage of contig 1 when aligned to the reference sequence: MFV LSFlaviV-A20-09. The X-axis and Y-axis indicates position and sequencing depth, respectively. (**B**) Maximum likelihood tree was plotted based on conserved amino acid sequences of viral polyprotein. The GenBank numbers of representative viruses were coded behind the virus name. cISFs (classical insect-specific flaviviruses)—purple, dISFs (dual-host affiliated insect-specific flaviviruses)—yellow, MBFVs (mosquito-borne flaviviruses)—red, TBFs (tick-borne flaviviruses)—cyan. The bootstrap value is 5000. The best-fit model was estimated as the LG + F + I + G4.

**Figure 4 viruses-14-01298-f004:**
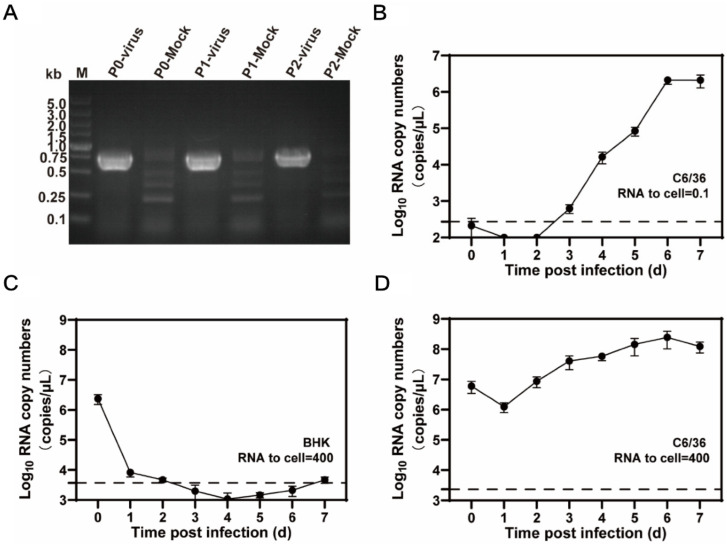
Rescue and characteristics of rescued MFV (rMFV) YN15-283-02 in vitro. (**A**) Detection of viral genome in cellular total RNA of P0 to P2. Total RNA from the infected and naïve cells was extracted and subjected to RT-PCR detection using the primers within the NS1 gene. The resulting RT-PCR products were resolved via 1% agarose gel electrophoresis. (**B**) Growth kinetics of rMFV YN15-283-02 in C6/36 cells. C6/36 cells were infected at a ratio of viral RNA copies to cell number of 0.1. Supernatants were collected at the indicated time points. (**C**,**D**) Growth kinetics of rMFV YN15-283-02 in BHK-21 (**C**) and C6/36 (**D**) cells at a high infection dose. The two cell lines were infected at a ratio of viral RNA copies to cell number of 400. Supernatants were collected at indicated time points. (**B**–**D**) Viral genome copies in the supernatant were quantified via qRT-PCR. Data represent mean ± SD from one representative experiment of the three independent experiments performed in duplicate.

**Figure 5 viruses-14-01298-f005:**
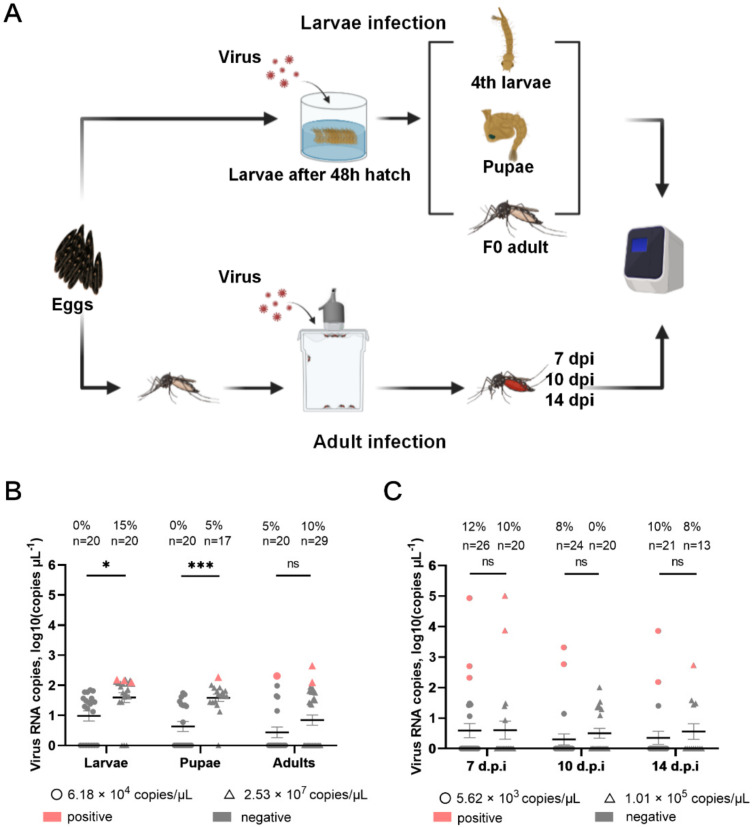
The result of mosquito infection through oral infection. (**A**) The process of mosquito infection was divided into larvae infection through feeding in water and adult infection through blood-feeding. (**B**) The larvae mosquito infection was detected via qRT−PCR. (**C**) The adult mosquito infection was detected via qRT−PCR. (**B**,**C**) The detection limit was 2.08 Log10 copies/μL (Ct = 35); *n* stands for the total number tested; the positive and negative samples are indicated in red and gray, respectively; *ns* (*p* > 0.05), * (*p* < 0.05), *** (*p* < 0.01).

**Table 1 viruses-14-01298-t001:** Contigs with significant BLASTn similarities to known virus in supernatant from midge homogenate-inoculated cell culture.

Contig ID	Length (nt)	Blastn to NCBI nt Database		
		Best Hit (*e*-Value < 10^−6^)	Coverage (%)	Identity (%)
Contig 1	10,486	Mosquito flavivirus (LSFlaviV-A20-09)	99	98.87
Contig 2, 3, 8, 12, and 13	1764, 1956, 1120, 1795, and 2857	Tibet orbivirus (Fengkai): S5, S4, S8, S6, S2	99, 100, 99, 88, 99	96.58, 97.34, 97.60, 97.35, and 97.18
Contig 4, 5, 6, 7, 9, 10, and 11	3939, 1065, 1330, 2746, 828, 396, and 452	Tibet orbivirus (DH13C120): S1, S9, S7, S3, S10, S2, S2	99,100, 80, 97, 99, 21, and 18	97.48, 98.03, 95.42, 100, 98.54, 98.82, and 98.82

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
