# Peer review of "In Vitro and In Vivo Characterization of a New Strain of Mosquito Flavivirus Derived from Culicoides"

_viruses, 2022, doi:10.3390/v14061298_

Round 1

Reviewer 1 Report

Huang et al submitted to viruses an interesting study focusing on biological characteristics of a new strain of Mosquito flavivirus derived from Culicoides both in vitro and in vivo. In this study, they detected and identified two viruses, flavivirus, and Orbivirus, from the midge homogenate inoculated C6/36 cultures using next-generation sequencing. Electron microscopy results clearly showed the presence of viral particles from two viruses. Then, the authors constructed a full-length cDNA infectious clone of a new strain of MFV, YN15-283-02, which is driven by a T7 promoter and they successfully rescued the viruses. Even though the authors could not isolate the virus from cell culture, they demonstrated the powerful application of infectious clone technology to get the rescued virus based on full genome sequencing information. With this development of infectious clone technology, it also can be potentially developed as the vaccine’s platform for mosquito-borne viruses. To further characterize the rescued virus, they found species specificity of rMFV YN15-283-02 infection in mosquito C6/36 cells and BHK cells. In vitro study showed that the rescued MFV YN15-283-02 has a low infection ability in Ae. Aegypti. Overall, this study was well designed and provided fundamental characterization of a strain of Mosquito flavivirus.

The following suggestions are to improve the MS.

  1. the writing of the MS needs to be improved.
  2. The author claim that they observed the viral particle by TEM but could not isolate the virus from cell culture. Did the authors try different available Mosquito cell lines for viral isolation? How many times in the passages they could detect the presence of viral genome by PCR in cell culture?
  3. In figure 4A, at least five passages need to be applied for the detection of the viral genome from total RNA.
  4. Since the author could not compare the replication capability of parental virus and rescued virus, it will be better to compare the replication of different passages of rescued virus (such as P1 vs P3 Vs P5), this will help to confirm the stability of the rescued virus.
  5. Could the authors make an explanation why the infection rate of adult infection at 10 d.p.i was 0% in the high dose group (Line 259)
  6. To highlight the potential importance of this study, a good discussion about the application of infectious clone technology for vaccine design will be helpful. Those papers can be cited for discussion, PMID: 34835160/ PMID: 27392429/ PMID: 31826984.

Reviewer 2 Report

In this article the authors describe a new Flavivirus from Culicoides collected in Yunnan, China. Unable to isolate the virus from a co-infection, they reconstructed its genome using high-throughput sequencing and then assembled an infectious clone by successively assembling the various parts of the genome. Their genome is 99% similar to a previously published strain (LSFlaviV-A20-09). The bioinformatics part is far too imprecise to be published as it is (See my comments line by line) and the discussion part lacks comparison between the results obtained with their strain and the strain (LSFlaviV-A20-09) or other flaviviruses concerning the Aedes infection.

Line 45 An endonuclease can't cleave a polyprotein. The cleavage of a polyprotein can be autocatalytic or carried out by cellular peptidases.

Line 107 Chapter too simplistic. Many methodological descriptions are omitted here although we are at a key stage of this article: obtaining the sequence of the virus.

Line 116 "homolog references were downloaded" with which criteria, which threshold, which tools ... ?

How the start and the stop of the RdRp region were choose?

“The best fit model was estimated” how?

Line 201 “6473 reads of them were closely matched to …” What does that mean ?

How were these reads selected? Did you map your reads to the reference?

if so with which mapper, which options?

The M&M speaks of a "trinity" assembler while the results chapter does not mention any assembly?

Line 205 Figure 3A is not understandable. I don't understand the "coverage" legend for the ordinates ; what’s could be 100% coverage ?. If the curve represents sequence similarity as I think it does, it must have been calculated from a sliding window. How long were these windows? were they overlapping?

Line 205 Figure 3B We don't have enough information on the sequences used in the MSA.

Which length ? The MSA should be provided as suppl data.

Line 295 The authors note a difference between their virus and the LSFlaviV-A20-09 strain in terms of creation of CPE on C6/36 cells but do not mention whether the LSFlaviV-A20-09 strain is capable of infecting Ae aegypti.

Round 2

Reviewer 1 Report

The manuscript can be accepted

Reviewer 2 Report

/